# Segmentation of Glottal Images from High-Speed Videoendoscopy Optimized by Synchronous Acoustic Recordings

**DOI:** 10.3390/s22051751

**Published:** 2022-02-23

**Authors:** Bartosz Kopczynski, Ewa Niebudek-Bogusz, Wioletta Pietruszewska, Pawel Strumillo

**Affiliations:** 1Institute of Electronics, Lodz University of Technology, 90-924 Lodz, Poland; bartosz.michal.k@gmail.com; 2Department of Otolaryngology, Head and Neck Oncology, Medical University of Lodz, 90-001 Lodz, Poland; ewa.niebudek-bogusz@umed.lodz.pl (E.N.-B.); wioletta.pietruszewska@umed.lodz.pl (W.P.)

**Keywords:** vocal disorders, laryngeal high-speed video, image segmentation, acoustic recordings of voice, signal processing, multimodal sensing

## Abstract

Laryngeal high-speed videoendoscopy (LHSV) is an imaging technique offering novel visualization quality of the vibratory activity of the vocal folds. However, in most image analysis methods, the interaction of the medical personnel and access to ground truth annotations are required to achieve accurate detection of vocal folds edges. In our fully automatic method, we combine video and acoustic data that are synchronously recorded during the laryngeal endoscopy. We show that the image segmentation algorithm of the glottal area can be optimized by matching the Fourier spectra of the pre-processed video and the spectra of the acoustic recording during the phonation of sustained vowel /i:/. We verify our method on a set of LHSV recordings taken from subjects with normophonic voice and patients with voice disorders due to glottal insufficiency. We show that the computed geometric indices of the glottal area make it possible to discriminate between normal and pathologic voices. The median of the Open Quotient and Minimal Relative Glottal Area values for healthy subjects were 0.69 and 0.06, respectively, while for dysphonic subjects were 1 and 0.35, respectively. We also validate these results using independent phoniatrician experts.

## 1. Introduction

Regular assessment of the health of the human voice is important for the accurate detection of voice disorders with varied etiology. Exposure to the risk factors of voice disorders is increasing in the contemporary world. It is estimated that about a third of workers in industrialized societies use voice as their main work tool. UK figures report that over five million workers are routinely affected by voice impairment, at an annual cost of around £200 million [1]. In recent decades, constant advancements in technology and virtualization of life have rendered voice crucial for communication, particularly in the case of individuals for whom it is a primary tool of trade and who are exposed to excessive vocal loading, e.g., actors, singers, coaches, teachers, call-center workers, etc. Professional voice users report to otolaryngological and phoniatric outpatient clinics with common problems. Due to voice overload, the vocal folds may be affected and deformed by pathological abnormalities causing malfunction of the entire speech apparatus [2]. Incorrect phonation caused by excessive muscular activity may lead to loss of voice. The most common effect of abnormal phonation (hyper-phonation) is pathological changes that appear in the form of nodules, polyps, and the weakening of the arytenoid or thyroarytenoid muscles [3].

Precise assessment of voice disorders with the aid of modern technology enables a structural and functional assessment of the larynx. Vibrations of the vocal folds play an essential role in voice production [4]. During the periodic oscillation of the vocal folds, the area between the vocal folds, called the glottal area, changes, which results in periodic interruption of the expiratory airflow through the glottis. Oscillation disturbance affects voice quality. Therefore, an accurate assessment method of vocal fold vibrations is crucial for the early diagnosis and treatment of various pathologies of the larynx [5]. Innovative instrumental methods are steadily gaining importance in otolaryngological and phoniatric studies of voice disorders [6].

Currently, the diagnosis of voice disorders can be facilitated by computer-based processing methods that enable the computation of many diagnostically meaningful parameters [7]. The most common diagnostic methods rely on acoustic voice measurements during sustained production of vocal sounds, termed phonation [8,9,10]. The parameters characterizing voice quality can be computed from microphone recordings of the produced voice [10,11], subglottal neck-surface accelerometer-based force measurements [12,13,14], or with an electroglottograph—an apparatus measuring the amount of electricity that passes through the larynx [15].

Recently, it has been stressed that direct visualization of laryngeal glottal structures and phonatory function in the clinical setting is essential to assess larynx pathologies. Therefore, in the last decades, an increasing number of scientific studies have reported new developments in advanced methods of digital processing and analysis of images of vibrating vocal folds [2,16,17]. There are three basic techniques of image collection: laryngovideostroboscopy, videokymography, and laryngeal high-speed videoendoscopy (LHSV). 

Videokymography is a high-speed imaging method depicting one horizontal line transverse to the glottis. The successively collected line of pixels stacked (from top to bottom) into a matrix is presented in real-time on a standard monitor revealing a graphical representation of the spatial position of the vocal folds over time [18,19]. 

The most common visualization method used in clinical practice is laryngovideostroboscopy (LVS) [11], although it does have significant limitations because visualizing a single vibration cycle of the vocal folds requires recordings taken from tens or hundreds of images from a sequence of consecutive cycles. Thus, vibration disorders of intermittent nature cannot be adequately detected, which has a detrimental effect on the quantitative analysis of LVS images. 

LHSV is the tool that provides the most precise insight into the function of the larynx during sound production. High-speed digital imaging of the oscillating vocal folds enables visualization of the true frame-by-frame movement of the vocal folds during sound production. However, this imaging technique requires the application of an advanced and expensive system that allows thousands of images to be recorded per second.

Recently, extensive research has been carried out in the field of a quantitative assessment of the glottal cycle using laryngeal high-speed digital imaging [16,20]. However, the main research problem encountered in these studies has been the development of automatic image analysis methods for segmenting the images of the larynx so that the boundaries of the vocal folds and consequently the glottal area could be reliably detected in each LHSV frame. If the segmentation process is inaccurate, the time- and size-related parameters characterizing the kinematics of the vibrating vocal folds will have little clinical relevance for the otolaryngologist and phoniatrician [21,22]. A detailed list of these parameters is provided in our previous study [11]. For a more in-depth review of related work, see Section 2.

In this work, we propose:an original method for automatic segmentation of the laryngoscopic images registered with LHSV based on the fusing of time-synchronized data modalities coming from the acoustic measurements of the produced voice and LHSV recordings.incorporation of the spectral domain data of the acoustic signal to control and optimize the segmentation algorithm of the LHSV images of the larynx during phonation.

We propose a novel approach to segmenting images of the moving objects to find other, non-medical applications. We demonstrate that it is effective in segmenting images of the vibrating vocal folds, and phoniatricians positively evaluated our results. The main advantage of our method is that it allows automatic segmentation of LHSV images without the need for trial and error in the search for optimal segmentation parameters.

The paper is structured as follows. In Section 2, we review recent studies related to the analysis of LHSV images. Section 3 describes the apparatus used to record LHSVs and introduces the medical cases of voice pathologies considered in this study. Section 4 explains the proposed method of automatic segmentation of LHSV images and presents the results that verify it in Section 5. The potential of the presented approach is appraised in the discussion in Section 6. Finally, overall conclusions from the presented study are formulated in Section 7.

## 2. Related Works

In recent years, a majority of the image analysis techniques of the vocal folds phonation process have relied on LHSV recordings [23,24,25] because the information from images of the oscillating vocal folds recorded at a rate of approximately 4000 frames per second (fps) provides a greater tracking precision of the movements of the vocal folds [26]. In particular, thanks to a very short image acquisition process, the problem of the movement of the laryngoscope with respect to the larynx, which tends to complicate algorithms for the analysis of the laryngostroboscopic images, is minimized. That is a great technological advancement compared to laryngovideostroboscopic techniques, which involve long image acquisition time, e.g., up to 20 s, and reconstruction of a single vibration cycle of the vocal folds from many consecutive vibration periods. Moreover, laryngovideostroboscopic techniques enable the visualization of real-time kinematics of transient vibration disruptions that tend to accompany some important voice disorders [27].

However, whichever image acquisition technique is applied, quantitative analysis of laryngovideoscopic images requires complex image analysis methods and additional adjustment, e.g., the settings of the parameters used in the segmentation algorithms need to be established to achieve reliable results [28]. In particular, the first step in the quantification of vocal vibration kinematics is a segmentation of the glottal area, i.e., the region between the vocal folds, in each consecutive image of an LHSV sequence. Should that segmentation step be flawed, any further parameters characterizing geometric and time-related parameters of the VF movement will be inaccurate. 

The development of a reliable image segmentation method of the larynx is a major challenge for automated computer algorithms for the following reasons [29]:a single view two-dimensional projection of a three-dimensional anatomic structure of the glottis is recorded; in particular, not all details of the elastic deformations of the vocal folds during the vibration cycle can be viewed,the point light source coming from the laryngoscope illuminates different anatomical regions of the glottis with nonuniform intensity,the position of the laryngoscope with respect to the glottis is different in each laryngeal examination, e.g., the distance and the viewing angle of the camera,the ground truth information about the glottal area can be collected only from subjective inspections and manual delineations of an expert doctor or a phoniatrician; for a very large number of images, the task is effort and time-consuming, and what is more, a special user interface needs to be developed to enable clinicians to precisely delineate the glottal area.

One of the most advanced approaches to the analysis of LHSV images was presented in [30]. The authors employed the Kalman filter to estimate the kinematic sequence of each of the vocal folds’ edges to predict the contact force during their collision. Researchers in [31], on the other hand, defined the region of interest (ROI) containing the image of the glottal area by analyzing the average intensity variations both in the columns and in the rows of the images.

In [32], a novel method was proposed for automatic glottis segmentation in endoscopic high-speed videos. ROI detection was done using the Fourier descriptors and a thresholding method combined with a level set algorithm, incorporating the prior glottis shape. The level set method is a numerical technique closely related to the active contour framework used to trace shapes of selected figures and identify dependencies among them based on the energy minimization criterion. Another advanced segmentation method utilizing the level set based curve evolution for vocal folds segmentation was proposed in [33]. However, the authors noted that the method required subjective parameter tuning and was unsuitable for fully automated analysis of the vocal fold movements during phonation.

It should be noted that most of the developed methods that have been proposed for segmentation of glottal images are designed for specific image recording conditions and work properly only for local databases of videos collected at institutions, hospitals, or health centers, and thus require manual validation (especially in the case of new registrations). Designing an algorithm that would yield satisfactory results for any given laryngoscopic video registration of the vocal folds during voice production (phonation) is a complex task. 

There is a group of image segmentation approaches that use continuity conditions derived from image sequences and extend the analysis into the time domain, e.g., by adapting a Geodesic Active Contour model defined in three dimensions and formulating continuous and variational energy minimization problems. The 3D surface is automatically evaluated through an algorithm optimizing the forces derived from the image and the surface shape (curvature and continuity), which minimizes the hypothetical energy functional [34]. Other methods adopt a Canny edge detector preceded by a 3D mean curvature filtering process [26]. 

Several methods have been proposed to validate the vocal folds image segmentation results with ground truth, based on calculating a metric of similarity between human- and machine-generated results [35]. The main problem related to objective evaluation is the necessity of generating ground truth, which is subjective and requires considerable time and effort [35].

A recent paper [36] tested different configurations of deep convolutional long-short-term memory networks were tested for automatic segmentation of the glottis and vocal folds in endoscopic LHSVs. The best-performing network was selected for extensive testing on a large set of LHSVs. Interestingly, the long-short-term memory architecture allowed the modeling of the spatial and temporal features of the vibrating vocal folds. This machine learning approach enabled fully automated quantification of the vibrations of the vocal folds. Nevertheless, the network required 13,000 LHSV frames to train the network. High segmentation precision was achieved, resulting in Dice coefficients values used for quantifying the segmentation results exceeding 0.85.

Other very recent work concerns the determination of the effect of incorporating features derived from vocal fold vibration transitions into acoustic boundary detection [37], comparative analysis of rapid videolaryngoscopy images and sound data [38], and a computer model for the study of unilateral vocal fold paralysis [39]. Interestingly, a method for detecting COVID-19 by analyzing vocal fold vibrations has also been proposed [40]. The presented literature review concludes that most image segmentation algorithms require a particular validation procedure to prove their accuracy. However, we have noted that a properly selected image segmentation technique combined with correlated acoustic analysis allows us to objectify the delineated contour of the vocal folds and provide a compliance parameter, which is crucial for quantitative image-based segmentation results of the glottal area. In our seminal work, we proposed such an image segmentation method [41]. The method is based on comparing the segmentation result with the synchronously collected acoustic registration during the patient’s phonation of the vowel /i:/. This paper expands on the first study and validates the results on a set of LHSV recordings for normophonic and dysphonic voices. Previous work was tested on only a few cases and did not include a broader discussion of the results and clinical interpretation by phoniatricians. In this study, we also fully automate this method by automatically detecting the glottal folds region of interest (RoI).

## 3. Materials and Recordings of LSHV

The laryngeal recordings were carried out at the Department of Otolaryngology, Head and Neck Oncology, Medical University of Lodz. 

Twenty-two subjects participated in the study, 7 males and 15 females (see Table 1). Eleven of the participants had normophonic voice (denoted N1–N11), whereas 11 were patients (denoted D1–D11) with voice disorders (dysphonia) caused by glottal insufficiency due to incomplete glottal closure. The age of the normophonic group (7 females and 4 males) ranged from 27 to 65 years, with a mean age of 46. The dysphonic group consisted of 8 females aged 26–64 (with the mean of 47 years) and 3 males aged 57–71 years (the mean = 65). Among the dysphonic patients, only the two oldest men experienced no professional vocal loading.

For all subjects, after a routine Ear, Nose, and Throat (ENT) examination, the imaging recordings of the larynx were performed using an LHSV system. In the normophonic patients, the LHSV examination showed no significant deviations in the regularity and symmetry of vocal folds vibrations, mucosal wave, and glottal closure (Figure 1). 

However, in three of the examined normophonic women, the imaging of the larynx revealed slightly incomplete glottal closure only in the 1/3 posterior part of the glottis, which did not affect their voice quality. In the dysphonic patients, disturbances of vocal fold vibrations and incomplete closure of the glottis during phonation were observed. The Glottal Closure Types (GTs) were described according to the guidelines of the Committee on Phoniatrics of the European Laryngological Society (ELS) [5], in the following way: type A is rectangle/longitudinal glottal closure, B—hourglass, C—triangle, D—V-shaped, and E—spindle-shaped. An illustration of these types of glottal closure is shown in Figure 2.

In the dysphonic subjects, the most commonly occurring abnormality was the spindle-shaped glottal closure. In 8 of the dysphonic patients, the spindle-shaped glottal gap along the entire membranaceous part of the glottis during the closed phase of the glottal cycle was observed (see, e.g., Figure 3). They complained of several voice-related problems: permanent hoarseness, vocal fatigue, and periodic voicelessness/aphonia. In one of the patients who reported periodic vocal fatigue, the longitudinal glottal closure (little incompleteness of glottal closure in the 1/2 posterior part of the glottis during the closed phase of the glottal cycle) was observed (Figure 4). One of the dysphonic subjects presented only a minimal spindle-shaped glottal gap in 1/3 middle part of the glottis (Figure 5).

The vocal fold function was assessed during sustained phonation of vowel /i:/ at a pitch and loudness comfortable for the subject. Simultaneously to the LHSV imaging, a synchronized acoustic recording of the voice produced during phonation was done. The recordings were repeated several times for each of the examined subjects.

The laryngeal images were recorded with a high-speed laryngeal camera from Diagnova Technologies with a 2/3-inch progressive CCD sensor with a camera shutter connected to an external microphone that synchronously recorded the acoustic wave generated by the vocal folds during the phonation. The image capture rate of the camera was 3150 images per second. The images were digitized at a resolution of 480 × 400 pixels. The inherent geometric lens distortions were corrected with calibration methods based on pixel coordinate remapping [35]. The light source was a 15 W laser with special spectral characteristics to achieve excellent visualization of the glottal tissue. The light from the illuminator was transferred to the endoscopic optics via an optical fiber. The camera was equipped with an electronically controlled lens allowing for manual or automatic image focusing. The Fiegert-Endotech ø12.4/7.2 endoscope used in the laryngeal recordings together with the assembled complete laryngeal high-speed system is shown in Figure 6a,b shows a diagram of how the laryngoscope is positioned in the larynx during the examination. Simultaneously to the LHSV recording, a synchronized acoustic recording of the voice produced during the sustained phonation of vowel /i:/ was done. The microphone we used for the voice recordings was an electret microphone MK602762PC featuring 20 Hz–16,000 Hz bandwidth. The relative distance between the microphone and the subject during the recordings was approximately 30 cm. The acoustic wave signal was sampled at a sampling rate of 22,050 Hz [36]. 

For the recording rate of the high-speed laryngeal images at 3150 frames per second, the camera captured approximately 200 images during one oscillation cycle of the VFs. This image frame rate was approximately 7 times slower than the sampling rate (22,050 Hz) of the acoustic signals, i.e., 7 acoustic audio samples were recorded during the acquisition of one LHSV image. In further analysis, the acoustic signal was down-sampled (as further explained in Section 4) to properly match its sampling rate to the image acquisition rate.

## 4. Automatic Segmentation Method of LHSV Images

The recorded LHSV RGB image sequences were converted to grayscale images using the standard formula [42]: Grayscale = 0.299R + 0.587G + 0.114B, where R, G, B are the red, green, and blue color components, respectively. Then, each image frame from the LHSV sequence was rotated so that the main axis along the glottal area was positioned vertically.

An important pre-processing step of the analysis of laryngeal images is to identify the region of interest (ROI), i.e., the region containing the vocal folds. We applied an efficient way of locating the ROI based on calculating the total image variation quantity as proposed in [31]. This quantity is obtained by calculating the sum of the absolute differences of image brightness in successive image frames. Thus, rapid changes in image brightness (e.g., due to moving vocal folds) will yield large values of this quantity. We define this quantity as the total variation map $TV\left(x,y\right)$ calculated over a sequence of frames as follows:(1)$$TV\left(x,y\right)=\overset{N-1}{\underset{t=0}{\sum }}\left|I\left(x,y,t+1\right)-I\left(x,y,t\right)\right|$$
where: $I\left(x,y,t\right)$—is the intensity function of the image at spatial coordinates *x*, *y*, and *t* denotes the frame index *t* = [0, 1, …, *N* − 1] of *N* analyzed images from the LHSV sequence. Points of *TV*(*x,y*) map assume large values for those image locations where there is a large variability of image brightness for consecutive images of an LHSV sequence. The map serves to locate the ROI for further image analysis. Figure 7 presents the obtained heat map based on the established ROI. 

The most important element of the quantitative assessment of the phonatory process is the automated localization of the vocal fold edges during voice production. When VF edges are correctly detected, a glottovibrogram, Glottal Area Waveform (GAW), Glottal Gap Waveform (GGW) can be constructed. These representations provide a complete characterization of the kinematics of the vocal folds boundaries. From these representations, numerous geometric parameters of the glottal area shape and its variation over time can be calculated. The definitions of the parameters used in this study can be found in Appendix A. Additionally, a more complete set of parameters characterizing glottal area geometry used to quantify other laryngeal pathologies is defined in our previous work [11]. 

The image processing pipeline is shown in Figure 8 and is as follows.

The color image of the glottis (Figure 9a) captured by the high-speed camera is converted into a greyscale image, then the region of interest is selected, and the edges of the vocal folds are detected (Figure 9b). Based on this, the glottal area, i.e., the space between the vocal folds, is determined (Figure 9c). The GAW is the signal representing instantaneous variations of the glottal area in time (Figure 9e). From the GAW, one can calculate geometric and time-related parameters characterizing the oscillation process of the VF, e.g., the minimum and maximum values of the glottal area [11].

The GGW is the signal representing instantaneous variations in the width of the gap between the VFs computed at predefined levels of the glottis. From the GGW, one can calculate the closing and opening periods of the VFs during a vibration cycle [11].

The glottovibrogram, on the other hand, is a spatio-temporal map illustrating time variations of the width of the glottal gap at different levels of the glottis. The glottovibrogram shown in Figure 9d depicts a map in which the columns represent time and rows correspond to the glottal width along the anterior-posterior length of the glottis. The instantaneous glottal gap width is represented by pixel brightness in the glottovibrogram map. 

In this work, we propose an automatic method for detecting VF edges based on the combined analysis of the data derived from LHSV recordings and synchronously recorded acoustic signals. We show that the segmentation algorithm can be optimized by the spectral data of the acoustic signal without the need to refer to ground truth information.

The underpinning idea of the method shown in Figure 10 is to pool candidate segmentations of the glottal images for a large set of segmentation parameters. Then, select the best segmentation result by finding the best match between the pool of Fourier amplitude spectra computed from the glottovibrograms and the Fourier amplitude spectrum computed for the synchronously recorded acoustic signal.

The applied segmentation method of the glottal image is based on a simple image thresholding method, as follows:(2)$$I_{O}\left(x,y\right)=\left\{\begin{array}{c} 0\;ifmed_{\propto }\left(x,y\right)-I\left(x,y\right)<\beta \\ 1\;ifmed_{\propto }\left(x,y\right)-I\left(x,y\right)\geq \beta \end{array}\right.$$
where:

*x*, *y*—pixel coordinates of the monochrome image,

*I*—image recorded during the phonation process,

*I_O_*—binary image containing the thresholding result,

*med_α_*(*x,y*)—median value computed at image coordinates *x, y* for pixels in a block size *α* × *α*,

*α*—the first segmentation parameter, i.e., the block size of the median filter,

*β*—the second segmentation parameter.

Parameter *α* specifies the block size of the median two-dimensional filter, and parameter *β* acts as a threshold value for the subtraction result between the image pixel *I*(*x,y*) and median filtered pixel *med*(*x,y*) at coordinates *x*, *y*. Note that the parameter *α* determines the strength of the median filter *med_α_*(*x,y*), i.e., the larger the filter window size (larger α), the stronger the smoothing effect the filter will have. Then, according to Equation (2), from this filtering result, the image content *I*(*x,y*) is subtracted. We can interpret this operation as removing the constant component from the image, computed for the image window size defined by parameter *α*. The remaining image data, i.e., devoid of the constant component, is thresholded at a level determined by the parameter *β.* The outcome of this segmentation method is a binary image consisting of pixels that are assigned values 0 (corresponding to the minimum pixel brightness) and 1 (corresponding to the maximum pixel brightness). 

This segmentation method is applied for a pool of segmentation parameters *α* and *β.* As a result, we obtain *M* = *α* × *β* segmentation results in Figure 10. Parameters *α*, *β* assume integer values in the range of [1, 255]. The task is to select from the *M* segmentation results the one that best fits the glottal area.

We propose the following multistep automatic procedure for selecting the best segmentation result for an LHSV recording consisting of *N* images of the glottis (refer to a graphical illustration of this method in Figure 10):

1.Compute a pool of *M* sequences of binary images; each sequence consists of *N* binary images obtained by segmenting LHSV images by applying the selected parameter combination (*α*, *β*) of the segmentation procedure as defined in Equation (1).2.For each of *M* sequences for the selected parameters (*α*, *β*), compute the glottovibrogram $g_{\alpha ,\beta }\left(t,l\right)$, where *t*—is the discrete time coordinate (the horizontal axis) and *l*—is the level along the glottal length (the vertical axis).3.For each of *M* glottovibrograms, compute the Fourier spectrum along *L* rows of the glottovibrogram and sum the results:(3)$$F_{\alpha ,\beta }\left(f\right)=\frac{1}{N}\overset{L-1}{\underset{l=0}{\sum }}\overset{N-1}{\underset{t=0}{\sum }}g_{\alpha ,\beta }\left(t,l\right)e^{-j\frac{2\pi t}{N}f}$$

where:

$g_{\alpha ,\beta }\left(t,l\right)$—the point of the glottovibrogram map computed for a parameter set (*α*, *β*),

$F_{\alpha ,\beta }\left(f\right)$—Fourier coefficients of the glottovibrogram,

f—frequency,

N—the number of analyzed consecutive LHSV images, 

L—the number of levels at which the glottal length is represented, i.e., the number of rows of the glottovibrogram.

4.Compute the Fourier spectrum of the acoustic recording *s*(*t*) performed synchronously with the LHSV recording:
(4)$$S\left(f\right)=\frac{1}{N}\overset{N-1}{\underset{t=0}{\sum }}s_{d}\left(t\right)e^{-j\frac{2\pi t}{N}f}$$

where:

$S\left(f\right)$—Fourier coefficients of the acoustic recording,

$s_{d}\left(t\right)$—downsampled (decimated) acoustic signal (as explained below),

f— frequency, 

N— the number of acoustic samples (after down-sampling).

Note that the down-sampling of the acoustic signal is necessary before the glottovibrogram Fourier spectra and the acoustic spectra computed in Equations (3) and (4) can be compared. The acoustic signal is down-sampled by a factor of 7, i.e., from the sampling rate of 22,050 to the sampling rate of 3150, which is equal to the acquisition frame rate of the LHSV sequence. Before down-sampling, the acoustic signal is low-pass filtered using 4-th order Butterworth filter with a cut-off frequency *f_c_* = 1500 Hz to meet the sampling theorem condition that the maximum frequency components of the sampled signal cannot exceed half of the sampling rate. 

5.For each combination of parameter values (*α*, *β*) compute the cost function $d_{\alpha ,\beta }$ to compare the modulus of the glottovibrogram spectra and the modulus of the acoustic spectra:(5)$$d_{\alpha ,\beta }=\overset{N/2-1}{\underset{f=0}{\sum }}\left|\left|F_{\alpha ,\beta }\left(f\right)\right|-\left|S\left(f\right)\right|\right|$$

where:

|·|—denotes the modulus of the Fourier coefficients.

6.Find a parameter set (α*, β*) for which the cost function is minimum:
(6)$$argmind_{\alpha ,\beta }=\left(\alpha^{*},\beta^{*}\right)$$7.Select the best segmentation result of the glottal image according to the criterion (6), obtained for parameters (*α*^*^, *β*^*^).

The values of the cost function $d_{\alpha ,\beta }$ computed for a set of parameters (*α*, *β*) are shown in Figure 10 in the form of a grayscale image where the value of the cost function is represented by pixel brightness. An asterisk denotes the minimum of the cost function. The example of the best fit of the Fourier amplitude spectra is shown at the bottom of Figure 10.

We should note that the proposed method involves a high computational cost due to the optimization process in which the best set of segmentation parameters (*α*^*^, *β*^*^) is selected. This optimization method requires the computation of *N* × *N* = 255 × 255 = 65,025 segmentations of each image frame from the LHSV recording. Then, for a series of segmented images, the corresponding glottovibrograms must be constructed. Their Fourier spectra have to be calculated. Then, these spectra have to be compared one by one with the spectrum of the recorded acoustic signal, and the best fit between them has to be selected. We estimate the proposed method’s computation time to segment a single LHSV recording consisting of 256 images to be approximately 5 min for a PC equipped with an Intel i7 processor. However, mapping the proposed algorithm, which consists of multiple independent computational threads, to the Graphical Processing Units (GPU) would significantly mitigate this shortcoming of the algorithm. 

The image processing and analysis algorithms and acoustic signal processing algorithms were developed in Python and C++ programming languages using open libraries, i.e., NumPy, SciPy, Matplotlib, and OpenCV. For time-critical methods (e.g., computing the median of a subset of pixel values), the C++ programming language was used to create Python bindings. We used the Spyder Integrated Development Environment for programming in Python.

## 5. Results

The proposed method was tested on the LHSV recordings collected from 11 individuals with normochromic voices and 11 with pathological voices, i.e., glottal insufficiency. During the LHSV recordings, the requirement was to record the voice signal simultaneously during phonation of vowel /i:/. Both the video and acoustic recordings were pre-processed according to the procedures described in Section 4 to make them suitable for computing the Fourier spectra, i.e., the pool of glottovibrograms was computed for a set of candidate segmentation parameters (*α*, *β*) and the acoustic recordings were down-sampled to match the sampling rate of the signal (*fs* = 22,050 Hz) to the frame rate of the LHSVs (*f_v_* = 3150 Hz). 

In Figure 11, we show an example segmentation results for the normophonic subject N2 obtained for six random selections of parameter values (*α*, *β*) and one segmentation obtained for a parameter set (*α*^*^, *β*^*^), i.e., that minimizes the cost function defined in Equation (5).

### 5.1. Phoniatrician Validation of the Obtained Results

The complexity of the vocal fold anatomical structure makes it difficult to provide objective ground-truth annotations that would enable quantitative evaluation of the established vocal fold edge positions during phonation. 

Our attempt to use a graphics tablet to delineate vocal fold boundaries on LHSV images was labor-intensive and not very precise. The drawn lines in many segments had to be corrected, and the result was not satisfactory in most cases. Thus, this method of obtaining ground truth from phoniatricians for annotation of vocal fold boundaries did not work. 

Therefore, a different approach using the capabilities of the proposed image segmentation method was used, in which a large pool of candidate segmentation was computed. Our automatic segmentation method searched for the minimum of the cost function defined by Equation (5) to determine the optimal segmentation result. We asked phoniatricians to perform a similar task on a preselected set of image segmentation results, i.e., to select, according to their clinical experience, the segmentation results that best match the vocal fold boundaries. Then we compared our results with the indications of phoniatricians. Below is a more detailed explanation of our approach to validating the segmentation results.

For each of the 22 examined LHSV recordings for both groups of individuals (normophonic subjects and patients with glottal insufficiency), we prepared a set of 60 different segmentation results obtained for different parameter values (*α*, *β*) where only one segmented image was obtained for the parameters (*α*^*^, *β*^*^), i.e., the one that was selected as the best according to the minimum condition of the cost function Equation (5). Then, for the set of 60 segmented images computed for each of the LHSV recordings, we asked two independent expert phoniatricians to select the best three image segmentation results corresponding to the best detection of the location of the vocal fold edges. The set of 60 segmented images of the glottis selected for evaluation was obtained for 60 pairs of segmentation parameters (*α*, *β*) selected from the cost function map (see sample map in the left panel of Figure 11). The coordinates of these parameters in the cost function form a matrix of 6 × 10 regularly spaced points in the rectangular neighborhood of the parameters (*α*^*^, *β*^*^) corresponding to the minimum of the cost function *d**_α_*_,_*_β_*.

Importantly, for each of the recordings, the phoniatricians’ selection of the best three segmentation results included the segmentation obtained for the parameters (*α*^*^, *β*^*^). Another notable observation is that the selection done by each of the phoniatricians differed very little, regardless of whether they were concerned for the normophonic subjects or patients with glottal insufficiency (see Figure 12, for an example of segmentation results selected by phoniatricians).

### 5.2. Calculation of Geometric and Time-Related Parameters for the Segmented LHSV Images

The designed algorithms described in this work make it possible to determine the position of VFs edges in terms of function minimization tasks. The obtained and validated segmentation results make it possible to compute several indices that quantitatively characterize the kinematics of the vocal fold vibrations (the definition of the indices is given in Appendix A). The presented solution is the basis for an objectified and quantitative analysis. The values of the computed indices for the examined subjects are summarized in Table 2.

The study confirms that it is possible to calculate quantitative parameters describing vocal fold vibratory characteristics based on the computer segmentation of LHSVs. The quotients Closing Quotient (CQ), Open Quotient (OQ), and Speed Quotient (SQ) were computed based on the obtained glottovibrograms, and their values depended directly on the accuracy of image segmentation. Table 2 presents the values of the CQ, OQ, SQ calculated for the middle part of the glottis. The rationale for taking that approach is that most of the dysphonic patients (subjects 9/11) presented with the largest incomplete glottal closure in the middle segment of the glottis, classified as type E: spindle-shaped GTs according to ELS. Thus, the segmentation results are essential for accurate quantification of the VF oscillations. Moreover, according to [42], vibrations in the medium segment of the glottis play a major role in normal voices. The pathology assessment at this point is the most important in glottal insufficiency (complete lack of closure at this position). Thus, OQ calculated in the medium segment of the glottis is a meaningful parameter for this type of voice.

Please also see the box-and-whisker plot in Figure 13, showing the spread of the calculated quotient values. The boxes are drawn from first quartile Q1 to third quartile Q3 with a horizontal line within the box to denote the median. Out of the calculated quotients, the CQ and OQ assumed significantly different values in the normophonic and the dysphonic group (Figure 13). Moreover, the median values of OQ and MRGA quotients for healthy subjects were 0.69 and 0.06, respectively, while for dysphonic subjects were 1 and 0.35, respectively. Nevertheless, the differences for all calculated quotients for normophonic and dysphonic subjects were significant. See the bottom row in Table 2 with *p*-values calculated by applying the non-parametric Mann–Whitney U test to the calculated quotients for normophonic and dysphonic subjects. Note that all *p*-values are less than 0.05. 

See Figure 14 illustrating clear discrimination of the two examined groups of subjects for MRGA, CQ, and OQ quotients. The collected data can be further used to build a larger database, e.g., from multiple phoniatric clinics, and apply machine learning algorithms [43] to discriminate normophonic and dysphonic patients robustly based on the calculated quotients.

The OQ assumed the highest values for the patients with the spindle-shaped glottal gap along the entire membranaceous part of the glottis. The OQ in those patients reached the value of 1.00, confirming that their vocal folds remained open throughout the phonation cycle. Similarly, for those subjects, the MRGA characterizing the ratio between minimum and maximum glottal area in the glottal cycle assumed large values (median 0.35), which confirmed a lack of the glottal closure. In the normophonic subjects, the MRGA reached small values (median 0.06), reflecting complete vocal fold closure along the entire length of the glottis.

## 6. Discussion

Computer image analysis techniques have brought about major advances in medical diagnosis based on quantified analysis of biomedical images of different modalities. In this respect, the analysis of biomedical images of moving tissue is particularly challenging. The human vocal folds vibrate with a frequency exceeding 200 Hz in the case of women. Real-time monitoring of this complex physiological phenomenon makes great demands on image recording systems. Recently, researchers’ interest in laryngeal high-speed recordings has gained on the previously popular stroboscopic recordings as they required longer recording times and suffered from the inability to reproduce irregular phonatory functions of the vocal folds. Thanks to thousands of images recorded per second, high-speed cameras offer real-time insight into the movement of the oscillating vocal folds. Determination of vocal fold vibrations during the phonatory function of the larynx is a crucial element in the diagnosis of the clinical type of voice disorders. The LHSV technique is an innovative diagnostic tool used to visualize larynx kinematics. It offers an unprecedented quality of real-time visualization of VF phonatory movement [44].

Nevertheless, the task of image segmentation, whose goal is to detect and track the edges of the vocal folds, remains a difficult computational problem. Many approaches have been recently proposed for solving this problem, with those involving deep neural networks trained on laryngeal images of healthy subjects and patients with voice disorders showing the greatest promise. Although very successful such approaches require tens of thousands of training examples of laryngeal images to achieve image segmentation precision comparable to manual segmentations [38]. However, as the authors of this paper conclude, comparing these results with other approaches is not possible due to the lack of a suitable reference data set. One of the drawbacks of machine learning methods is that the results they produce lack explanatory power, and the segmentation decisions are hidden within the deep structure of the neural network. Nevertheless, advances in machine learning techniques towards explaining neural network decisions are ongoing and can certainly offer powerful tools for image recognition with explanatory features.

We have proposed a novel approach in which we control the image segmentation algorithm with data derived from acoustic recordings collected synchronously with the video capture of the phonatory process. It needs to be noted that the recorded acoustic signal is filtered by the vocal tract [45,46,47,48] and does not directly reflect the mechanical oscillation of the vocal folds. However, owing to the Fourier spectral representation, the fundamental frequency of the vibrations can be clearly outlined. During the phonation of vowel /i:/, the acoustic signal after computing its amplitude Fourier spectrum consists of the fundamental frequency and formants characterizing the acoustic properties of the vocal tract (as shown in Figure 10). It is worth noting that the harmonic corresponding dominates such a spectrum to the fundamental frequency. 

The basis of our method is the assumption that the best fit between the amplitude Fourier spectra of the acoustic signal and the spectra derived from the glottovibrogram will occur when the sequence of segmented images reflecting vocal folds movements is represented by the harmonic identical to the frequency of vibration of the vocal folds, i.e., the fundamental frequency. In the case of dominance of other frequencies in the segmented images, i.e., different from the fundamental frequency, large values of the cost function (5) were obtained, indicating incorrect segmentation of the vocal folds, i.e., detection of laryngeal anatomical structures that do not represent the movement of the vocal fold edges.

We recognize that the study is not without its limitations: the method has only been tested not on a large number of recordings, and only one type of voice disorder involving glottal insufficiency was considered in the study. However, the video material was carefully selected under the supervision of phoniatricians, who selected the most important and representative cases for our study. At present, due to the limitations of COVID, we cannot collect video material on a larger scale that could include representative groups, particularly in terms of gender, age, and health status.

Moreover, it should be noted that it is possible to develop image segmentation methods other than ours that might give even better segmentation results. Our primary intention was rather to show the potential of our original approach to the problem of segmenting images of moving objects for those cases where other sources of data of different modalities are available and can be used to optimize the image segmentation process.

In subjects with voice disorders, impairments in vocal fold oscillations affect the acoustic quality of their voice. It should be noted that there are three main vocal fold dynamical features that foster normophonic/euphonic voice: vocal fold oscillations are assumed to be: (1) symmetric, (2) periodic, and (3) exhibit a closed state during oscillations [6]. Incomplete glottal closure during the phonatory function of the larynx is associated with vocal fatigue and a breathy voice. However, it is assumed that slightly insufficient dorsal glottal closure should be regarded as normal, particularly in women [49]. The study confirms that three of the examined women with normophonic voice observed incomplete glottal closure in the 1/3 posterior part of the glottis with no effect on voice quality. Other types of glottal insufficiency were considered pathological. In the examined dysphonic subjects, the spindle-shaped glottal closure (type E according to the guidelines of ELS) was observed the most frequently (9 patients). Its distinctive features were the bowed shape of the vocal fold edges and a lack of glottal closure in the inter-membranaceous part of the glottis, resulting from the asthenic or atrophied vocal muscles and mucosa in the vocal folds. Such structural modifications lead to increased glottal air leakage and a breathy, weak voice. Professional vocal loading and aging (presbyopia) are considered to be the most frequent factors predisposing to this kind of glottic dysfunction [49,50,51]. Our results concur with their study. The patients in our study who experienced the following voice symptoms: vocal fatigue, weakness of voice, or voicelessness, mainly included professional voice users with long-term vocal loading. In turn, the two oldest male participants with no experience of professional vocal loading had clinically diagnosed presbyphonia. 

Furthermore, the application of LHSV enabled the determination of the parameters OQ, CQ, SQ, and MRGA from the variations of the GGW in consecutive LHSV images. These parameters, characterizing incomplete glottal closure, have been found meaningful in assessing vocal fold vibrations, particularly vocal apparatus’ insufficiency, as reported in recent studies [52]. Such evaluations are important in clinical voice assessment, e.g., in objective diagnosis and monitoring the results of an administered therapy [53]. Determination of the computed parameters makes it possible to parametrize dysfunctions of the glottis, including asthenia of the internal muscles of the larynx affecting vibrations of the VFs. One of the most relevant indexes is the OQ representing the duration of the open phase in relation to the total glottal cycle. The OQ is considered a good measure for comparing normophonic subjects with patients suffering from glottal insufficiency. 

In [54], it was reported that in 96% of the patients with occupational voice disorders, the value of the OQ was on average 0.98, while the mean OQ value in the normophonic subjects took the value of 0.68. Moreover, in the dysphonic subjects, the MRGA, i.e., the ratio between the minimum and the maximum glottal area in the glottal cycle, assumed higher values than in the normophonic group (mean value 0.35 vs. 0.09) and quantified the incompleteness of the glottal closure (glottal gap). These results are consistent with our study and our earlier study [11] conducted on another group of patients. Analysis of videolaryngostroboscopic images was used for characterizing incomplete glottal closure. In another study [12], which used a high-speed video system to collect data from healthy subjects only, the mean value of OQ was 0.66 ± 0.14 in females and 0.56 ± 0.1 in males. These results again are consistent with our study. Similarly, the results for the mean value of SQ = 0.85 ± 0.21 obtained for the healthy female subjects are consistent with our results SQ = 0.77 ± 0.21. Also, in [55], in line with the studies discussed above, an elevated mean value of the OQ obtained for the patients with voice disorders was OQ = 0.84 ± 0.16 compared to healthy subjects OQ = 0.84 ± 0.16. Interestingly, and in contrast to the reviewed studies, in a recent work reporting an open platform for laryngeal high-speed videoendoscopy [29], very high values of the OQ were obtained for healthy subjects taking a mean value of 0.998. However, it was noted in [56] that singers might develop a special mechanism for voice production, the so-called laryngeal mechanism, in which OQ can reach high values exceeding 0.9. 

However, it should be noted that comparisons of different studies using LHSV imaging to quantify the vibration of the vocal folds should be made with caution [57]. In a recent work [7], which provided a comprehensive review of the computer methods for quantifying vocal folds vibration, the authors concluded that it is difficult to compare the effectiveness of different methods due to the lack of publicly available databases designed for benchmarking different laryngeal image analysis methods. This is because these studies use different laryngeal image datasets, different image acquisition equipment, different assessment methods, and individual performance metrics. Meaningful comparisons between different studies require publicly available datasets and establishing a set of guidelines, preferably developed by experts from different health centers. Finally, it is worth noting a novel computational approach for spatial segmentation of high-speed laryngeal videoendoscopy images in a connected speech presented in [58]. This approach aims to develop an LHSV-based measurement of the vibratory characteristics of the vocal folds based on natural speech production, as opposed to the traditionally used phonation examination protocol.

## 7. Conclusions

In this work, we have shown that it is possible to track the location of the edges of the vibrating vocal folds in LHSVs in a way that does not require manual validation or intervention by medical personnel. In particular, we have demonstrated that image segmentation techniques can be optimized by utilizing data derived from synchronously collected acoustic recordings during sustained phonation of vowel /i:/. By transforming the glottovibrogram to the frequency domain and mapping it onto a one-dimensional spectrum, we could compare it directly to the spectrum of the acoustic signal and build a relevant cost function. The minimum of the cost function was the criterion by which the best segmentation results were identified. Independent otolaryngology-phoniatrics experts successfully validated these results. 

We would like to strongly emphasize that the main value of our contribution, beyond the segmentation method itself, is the automatic technique for optimizing image segmentation methods of video images of the moving vocal folds where segmentation parameters can be defined, e.g., threshold value, size of the filtered neighborhood, etc. 

It is important to note that most of the developed methods for segmentation laryngeal images require manual tuning of many parameters to obtain acceptable image segmentation results. Our method allows automatic segmentation of LHSV images without trial and error in searching for optimal segmentation parameters. We hope that the proposed approach, which considers acoustic modality, may inspire other researchers to use this image segmentation technique.

The proposed method of segmentation of LHSVs enabled automated tracking of the vocal fold edges during phonation. On that basis, we computed the corresponding glottovibrograms and GGWs that allowed us to further calculate a number of indices quantifying pathological changes in the phonatory process. The current findings support the use of this analysis method in clinical practice, which promotes LHSV as a reliable laryngeal imaging tool. The calculated indices will allow clinicians to provide reliable measures to objectively assess laryngeal phonatory function. Quantitative assessment of vocal fold vibratory disturbances characteristic for the glottal insufficiency may improve the diagnosis of occupational voice disorders or presbyphonia. In recent decades these laryngeal diseases have attracted clinical attention due to the increasing number of professional voice users and the aging population in the modern world.

As indicated in the conclusion section, the proposed optimization procedure for the image segmentation algorithm involves a high computational cost that can, however, be mitigated by mapping the computations to GPU hardware.

Finally, we hope that the proposed method can trigger studies that will follow the proposed path and further explore the approach in which the fusion of data from LHSVs and acoustic recordings is used to optimize image analysis techniques aiding clinicians in the diagnosis and quantification of voice disorders. 

## Figures and Tables

**Figure 1 sensors-22-01751-f001:**
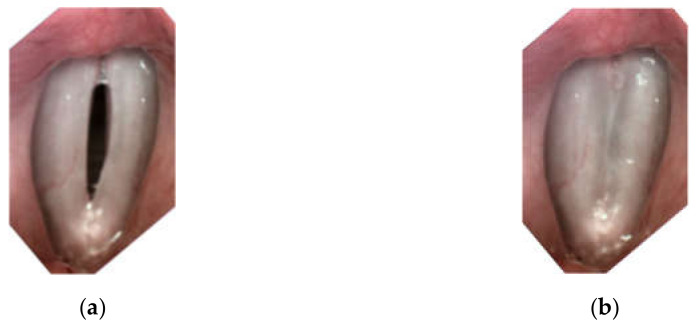
Images of the glottis for the normophonic subject N3 for the maximum opening (**a**) and maximum closing (**b**) of the vocal folds correspondingly.

**Figure 2 sensors-22-01751-f002:**
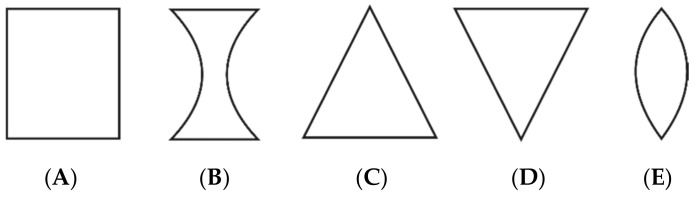
Classification of glottal closure types: (**A**) rectangle/longitudinal, (**B**) hourglass, (**C**) triangle, (**D**) V-shaped, (**E**) spindle-shaped.

**Figure 3 sensors-22-01751-f003:**
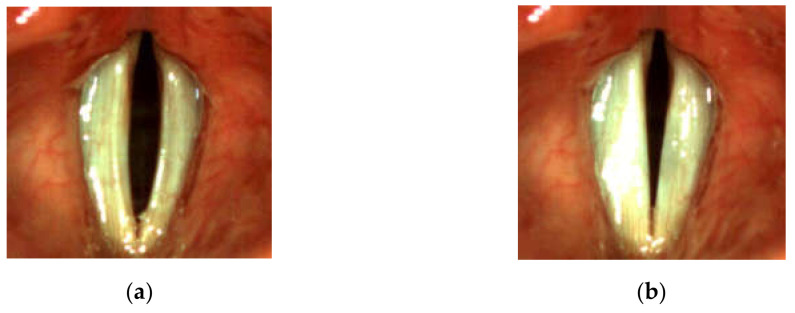
Images of the glottis for dysphonic patient D8 with severe glottal insufficiency for the maximum opening (**a**) and maximum closing (**b**) of the vocal folds correspondingly.

**Figure 4 sensors-22-01751-f004:**
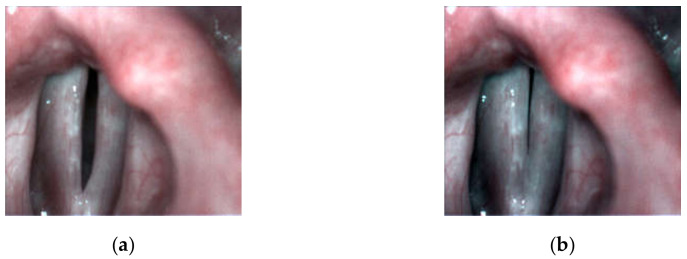
Images of the glottis for dysphonic patient D5 with longitudinal glottal insufficiency for the maximum opening (**a**) and maximum closing (**b**) of the vocal folds correspondingly.

**Figure 5 sensors-22-01751-f005:**
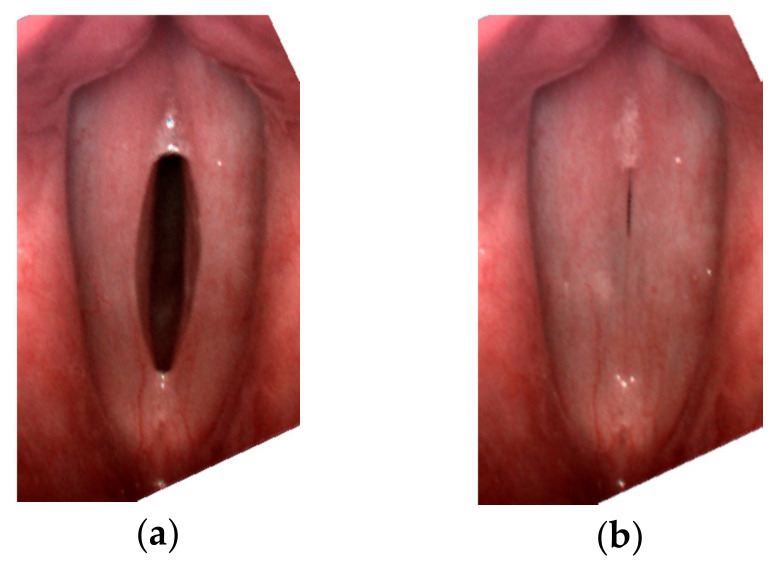
Images of the glottis for dysphonic patient D4 with minimal spindle-shaped glottal insufficiency for the maximum opening (**a**) and maximum closing (**b**) of the vocal folds correspondingly.

**Figure 6 sensors-22-01751-f006:**
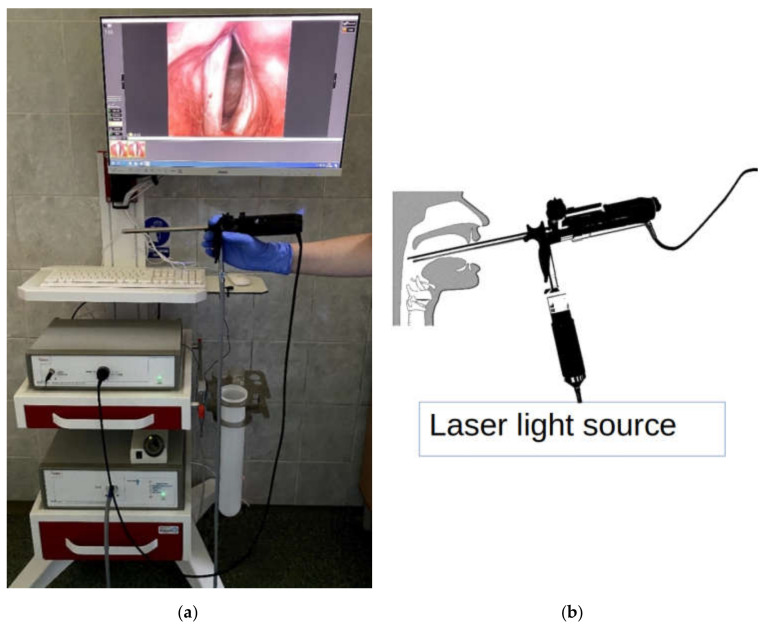
Photograph of the LHSV recoding system with the 70-degree rigid scope, attached light source, and a microphone. The box on the lowest shelf of the rack is the endoscope’s light source, and the box on the middle shelf is a high-speed camera offering acquisition of up to 4000 images per second (**a**), a diagram showing the position of the laryngoscope during laryngeal examination (**b**).

**Figure 7 sensors-22-01751-f007:**
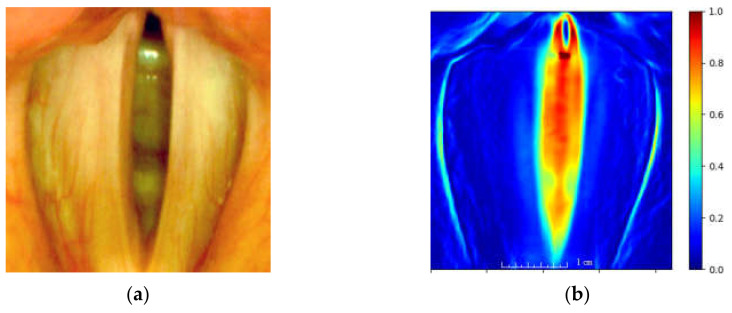
The image of the glottis of a normophonic subject (**a**) and the corresponding total variation image (**b**), as defined in Equation (1), represented as a heat map (the larger the variation, the warmer the color of the map).

**Figure 8 sensors-22-01751-f008:**
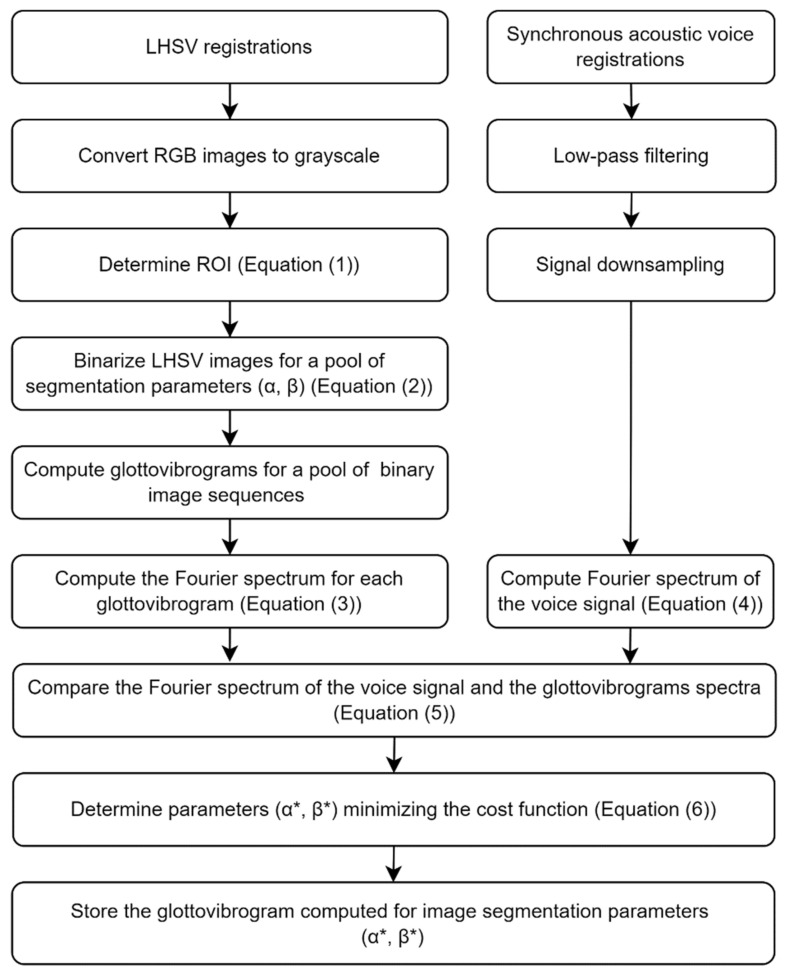
The processing pipeline of recorded LHSV images and synchronously recorded voice signal during sustained phonation of vowel /i:/.

**Figure 9 sensors-22-01751-f009:**
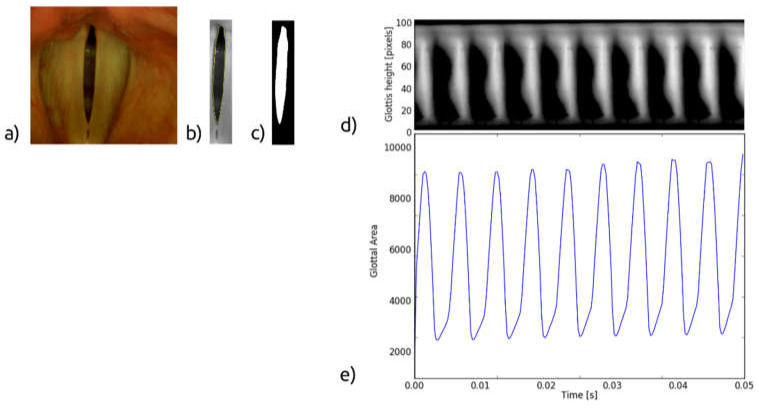
Representations of the LHSV image: (**a**) laryngeal image of the glottis, (**b**) detected contour of the glottal boundary, (**c**) glottal area, (**d**) the glottovibrogram, (**e**) the glottal area waveform.

**Figure 10 sensors-22-01751-f010:**
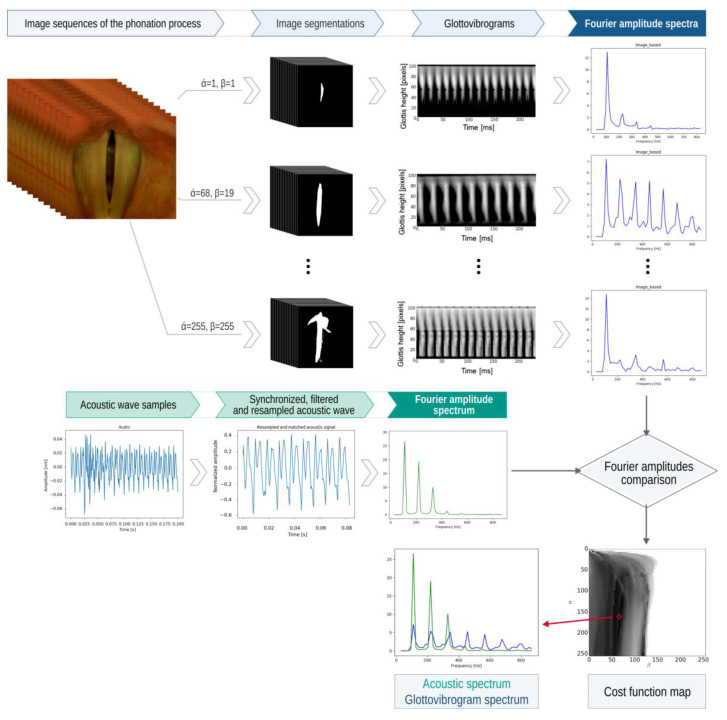
Diagram explaining the designed method of the segmentation of glottal images where, in the search for the best segmentation results, the Fourier spectra derived from the pool of segmented LHSVs are compared to the Fourier spectra of the acoustic recording.

**Figure 11 sensors-22-01751-f011:**
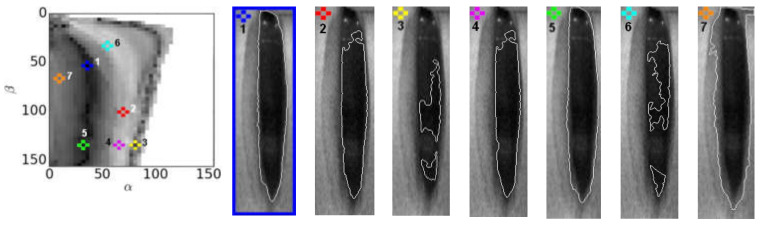
Plot of the cost function map *d**_α_**_,_**_β_**_,_* (left panel) and example image segmentation results (right panel) obtained for the normophonic subject N2. The segmentation results are obtained for parameters (*α*, *β*) and assigned different numbers in the cost function plot. The best segmentation result is shown in a thick box on the left side of the right panel and marked with the number 1.

**Figure 12 sensors-22-01751-f012:**
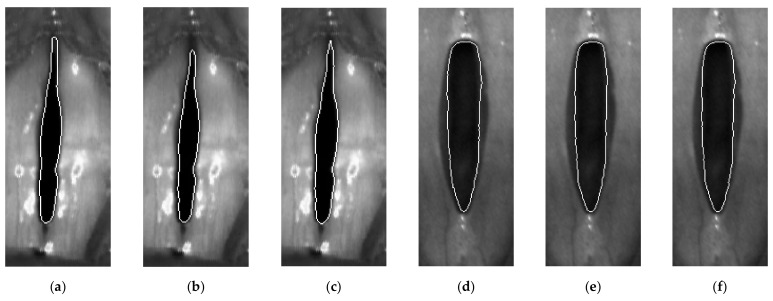
Example segmentations of the glottic images selected by the phoniatricians: images (**a**–**c**) are for the normophonic subject N10; (**d**–**f**) is for the patient I5 with glottal insufficiency; images (**a**) and (**d**) are the results obtained for the optimum segmentation parameter set (*α*^*^, *β*^*^) minimizing cost function (5).

**Figure 13 sensors-22-01751-f013:**
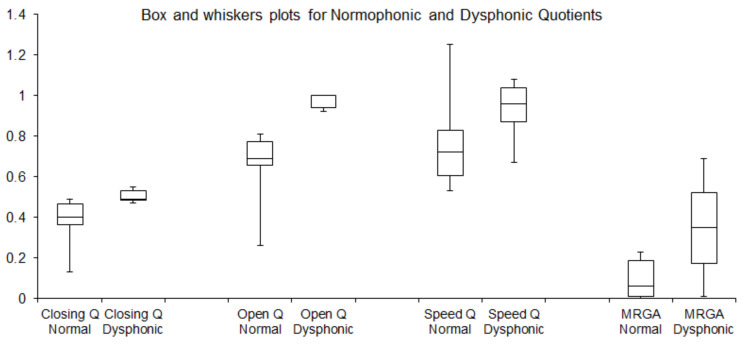
Box-and-whisker plots of calculated quotients for normophonic and dysphonic subjects. The upper and lower boundaries of the boxes indicate first quartile Q1 to third quartile Q3, respectively, while the boundary of the lower whisker denotes the minim value in the data set and the upper whisker boundary denotes the maximum value in the data set.

**Figure 14 sensors-22-01751-f014:**
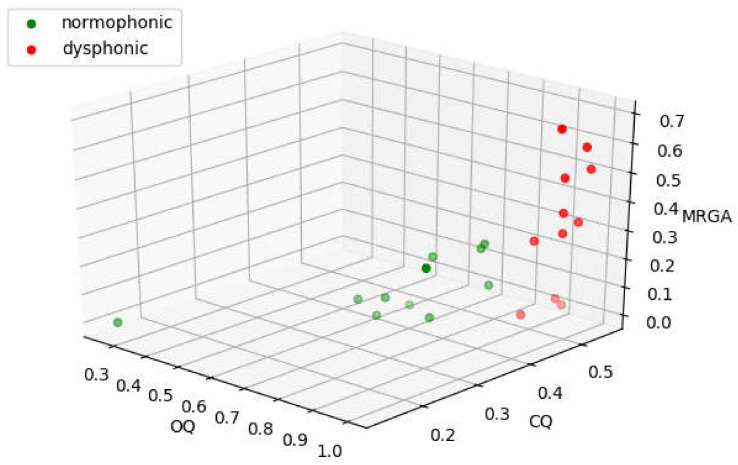
3D plot for indices MRGA, OQ, CQ illustrating good discrimination of the normophonic subjects (green dots) and patients with glottal insufficiency (red dots).

**Table 1 sensors-22-01751-t001:** Summary of patients participating in the study by gender and normophonic/dysphonic subjects.

	Patients	Normophonic	Dysphonic
Males	7	4	3
Females	15	7	8
Total	22	11	11

**Table 2 sensors-22-01751-t002:** Geometric and time-related parameters for the segmented LHSV images in normophonic subjects and dysphonic subjects with glottal insufficiency.

	Patient Number	Closing Quotient	Open Quotient	Speed Quotient	MRGA ^1^
Normophonic	N1	0.36	0.81	1.25	0.23
	N2	0.36	0.61	0.72	0.05
	N3	0.37	0.65	0.76	0.00
	N4	0.48	0.81	0.69	0.09
	N5	0.42	0.67	0.62	0.01
	N6	0.49	0.78	0.57	0.22
	N7	0.40	0.76	0.90	0.01
	N8	0.13	0.26	1.08	0.00
	N9	0.45	0.69	0.53	0.17
	N10	0.38	0.66	0.74	0.06
	N11	0.49	0.77	0.59	0.20
Dysphonic	D1	0.49	1	1.04	0.33
	D2	0.53	1	0.89	0.60
	D3	0.54	1	0.85	0.52
	D4	0.49	1	1.04	0.52
	D5	0.55	0.92	0.67	0.01
	D6	0.47	0.92	0.96	0.03
	D7	0.47	0.95	1.02	0.30
	D8	0.48	1	1.08	0.69
	D9	0.49	1	1.04	0.40
	D10	0.52	1	0.92	0.35
	D11	0.53	0.93	0.77	0.05
*p*-values		1.3 × 10^−3^	7.1 × 10^−5^	0.04	0.01

^1^ Minimal Relative Glottal Area.

## Data Availability

The acoustic recordings and segmentation results of the LHSV images reported in this work are available at: https://tulodz-my.sharepoint.com/:f:/g/personal/pawel_strumillo_p_lodz_pl/Eni2ELdrtmBOh_ez4ul-asIBmaZr_5pcHjaYjZH8R5b_YA?e=d1OCwL (last accessed on 20 December 2021).

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
