# Peer review of "Segmentation of Glottal Images from High-Speed Videoendoscopy Optimized by Synchronous Acoustic Recordings"

_sensors, 2022, doi:10.3390/s22051751_

Round 1
Reviewer 1 Report
The study regards important issue of objectivisation of vocal folds vibration parameters including glottal insufficiency. Of crucial importance is objective parametrization of the lack of glottal closure resulting from the dysfunction of particular muscle groups within larynx, which is possible due to the assessment of Open Quotient value in anterior, middle and posterior segment of the glottis. The technique used in this study is of small precision and excludes aforementioned assessment. In the literature of the subject, researches precisely determining the level of glottal insufficicency using high speed camera are present. The study submitted for the review is of low novelty and the methotodology used is unprecise. The weakness of this project is its small study group (11 subjects) which underestimates credibility of the results and enables to draw adequate conclusions.
In my opinion this study should not be published.
Author Response
Reviewer1:
Dear Reviewer, thank you very much for reviewing our article. We thank you for highlighting the importance of the study, which aims at clinical aspects, i.e., diagnosis of dysfunction of specific muscle groups within the larynx. Below are our point-by-point responses to the raised issues.
The study regards important issue of objectivisation of vocal folds vibration parameters including glottal insufficiency. Of crucial importance is objective parametrization of the lack of glottal closure resulting from the dysfunction of particular muscle groups within larynx, which is possible due to the assessment of Open Quotient value in anterior, middle and posterior segment of the glottis. The technique used in this study is of small precision and excludes aforementioned assessment. In the literature of the subject, researches precisely determining the level of glottal insufficicency using high speed camera are present
According to the work:
Osma-Ruiz, V.; Godino-Llorente, J.I.; Saenz-Lechon, N.; Fraile, R. Segmentation of the glottal space from laryngeal images using the watershed transform. Computerized Medical Imaging and Graphics 2008; vol 32, no.3, 193–201, doi.org/10.1016/j.compmedimag.2007.12.003
vibrations in the medium segment of the glottis play a major role in normal voices. The assessment of the pathology at this point is the most important in glottal insufficiency (complete lack of closure at this position). Thus, OQ calculated in the medium segment of the glottis is a meaningful parameter for this type of voice.
We added this information in the manuscript in a paragraph after Table 2.
Please note that we have shown statistical significance of the obtained results with p-values not exceeding 0.004.
The study submitted for the review is of low novelty and the methotodology used is unprecise. The weakness of this project is its small study group (11 subjects) which underestimates credibility of the results and enables to draw adequate conclusions.
In my opinion this study should not be published
We would like to kindly point out that we consider our method to be novel because, to the best of our knowledge, it is the first imaging segmentation method of the glottal area from the HSV laryngeal images that proposes an automatic optimization of the imaging segmentation process based on data derived from synchronously recorded voice signal (during the phonation of the vowel /i:/ of the studied patients).
A complete pool of segmentation results and acoustic recordings is available at the link, the best of which were judged by phoniatricians to be of high precision:
https://tulodz-my.sharepoint.com/:f:/g/personal/pawel_strumillo_p_lodz_pl/Eni2ELdrtmBOh_ez4ul-asIBmaZr_5pcHjaYjZH8R5b_YA?e=d1OCwL
We would like to emphasize that the main objective of our manuscript was to focus on the technical aspects (i.e. relevant to the scope of the Sensors journal) of our original method for image segmentation of the glottal area. We plan to collect more recordings (currently threatened by Covid pandemic) for a larger study group of healthy subjects and patients with various voice disorders. We plan to publish the results of this future study in a clinical journal, with major medical input from collaborating phoniatricians.
Please note, that the study group includes 22 subjects, i.e. 11 with glottal insufficiency and 11 with homophonic voices.
We believe these explanations will provide a more favourable view of our work

Reviewer 2 Report
General comment:
This work deals with the development of a methodology to combine and syncrhonize the LHSV data and acoustic recording during larygneal endoscopy.
The subject is interesting, the problem faced is challenging and the proposed solution can be of interest to the readership. Some aspects has to be clarified and few critical points must be addressed.
Specific comments throughout the paper:
Abstract
C1: Line 21: OQ and MRGA are defined as acronyms, but not re-used in the abstract. Please, define them in the text.
The abstract legnth should not be higher than 200 words.
C2: Lines 21-23: Missing spaces for the standard deviations. I suggest to remove the bracket parentheses.
C3: Keywords: The selected index terms are very "clinical", not strictly related to the Sensors journal. I suggest the authors to consider to provide some technical, more related to the signal and image processing, terms. The visibility of your work is important for you, your work, but also for the journal. Sensors is not a clinical, medical journal, even though retains an interdisciplinary character. The engineering, computer science and hardware aspects must be evident.
1. Introduction
C4: Line 30-31: Please be quantiative and provide some statistics and number for this statement, as well as reference(s).
C5: Line 51: Extra space.
C6: Lines 59-65 and Lines 81-82: It seems that the authors are not addressing directly the state of the art (however Sect. 2 is provided), therefore, I suggest to insert some sentence to explicitely state that the comparison with literature works is performed but in the following.
This would help the readers to navigate your work and find easily the useful information.
2. Related work
C7: Please revise the title, use the plural.
C8: Line 113: do not clamp "approx." use the entire word.
C9: Lines 111-123: I suggest the authors to consider to provide a figure which sketches the diagnostic procedure, as well as illustrate the problems and solution. This sort of graphical abstract can provide to non-technical readers (interested in re-adapting your methodology to similar but not identical problems) an immediate idea of your aim and scope, of your method and solution, while elucidating the LHSV details.
C10: Lines 124-127 and Lines 132-145: Unsupported statement. Missing reference. I suggest:
10.1109/JBHI.2014.2374975
10.1038/s41598-021-93149-0
C11: Lines 191-200: The authors reported their previous work [34], but did not highlight the differences with the submitted manuscript. Please, revise this part to better stress the novelty.
3. Materials and recordings of LSHV
C12: Lines 204-210: The authors should better summarize and organize the database to present it in a clear way. I suggest to use a table. It can be provided in the supplementary material and therefore a list of all subjects can be provided.
C13: Lines 220-222: I suggest to provide an image or a table (containing sketches) for this taxonomy.
C14: Line 247: Why red color? Please revise.
C15: Line 257: Use "W" for "-watt".
C16: Line 261: Please use the correct diameter symbols.
C17: Line 267: Please provide producer and some technical details for the mircophone.
4. Automatic segmentation method of LHSV images
C18: Line 284: Provide a reference.
C19: Lines 288-290: The authors are citing [39], but it would better, for the sake of clarity and to ensure a perfect reproducibility, to provide a brief summary of the procedure.
C20: Thanks for providing the explanation of Eq. (1) and Fig. 6. The colorbar for Fig. 6b is missing. Please fix.
C21: Lines 301-306: Why only three features were selected? Why only these? Please provide a justification or reference(s). I suggest to provide the mathematical definition or the geometric/pictorial representation in a dedicated figure.
C22: Lines 307-314: I suggest to provide the pipeline as a worflow diagram, also.
C23: Eq. (2): Please revise this part, the usage of the parameter alpha is not clear and some readers may loose where it enters.
C24: Line 363: Please revise the way the range is provided, use "[1,255]".
C25: Line 400: What "appropriate" means? Please provide the information about the filter (e.g., order, etc.).
C26: The coding language or the softwares used for processing the images and audio data is not provided. Please, provide all the methodological details.
5. Results
C27: Line 449: Revise "Eq.". Check elsewhere.
5.1. Phoniatrician validation of the obtained results
C28: Lines 459-4467: The validation is not clearly described. Did the authors performed a direct comparison of the segmentation perimeters or segmented areas derived automatically and by the experts? A suitable error metrics can be computed and a quantitative discussion can be provided. A deep and coherent discussion is in order. The validation of your approach is crucial.
5.2 Calculation of geometric and time-related parameters for the segmented LHSV images
C29: The authors can comment on the fact that the findings from Fig. 11 cna be relevant for regression analysis or some machine leraning methods (e.g., SVM).
6. Discussion
C30: The discussion section is of good quality.
7. Conclusions
C31: Conclusions are fine, even though this section is quite a summary.
Reviewer 3 Report
The primary advantage of this paper is the simultaneous recording of video and sound data during laryngeal endoscopy. We demonstrate how the glottal area image segmentation algorithm can be optimized by matching the Fourier spectra of the pre-processed video with the acoustic recording during vowel /i:/ phonation.
However, the material provided for the literature review is out of date. As a result, I propose including works that are more recent. Additionally, authors should monitor new publications on the web of sciences in order to credit relevant works produced between 2021 and 2022.
The abstract is effectively written; however, the writers could stress the contributions in the introduction section, primarily in terms of points, to ensure that readers comprehend.
While some may claim that the data is insufficient, it is not always possible to acquire enormous amounts of data.
However, I believe that this paper has potential, at least from a data science/artificial intelligence standpoint. The authors require further time to polish their article.
Round 2
Reviewer 1 Report
Author’s explanations are not satisfying for me. In my opinion, the manuscript should be rejected for publications what I have already stated in my previous review.
Author Response
Dear Reviewer,
as per your suggestion, we provided a citation [43], which demonstrates that the assessment of vocal fold vibration in the middle segment is the most important in glottal insufficiency.
In addition, we double-checked the statistical significance of the results using the non-parametric Mann-Whitney U test, obtaining p values less than 0.04.
Reviewer 2 Report
Thank you for your replies.
The authors explained to me the difference of this manuscript with their previous work [34] (C11), but they didn't account for this difference and the novelty in the modified version of the manuscript.
C12: Thank you for providing these data.
C25, C26: Thanks for providing these methodological details useful for the reproducibility.
C28: Thank you for addressing this rather crucial point.
Author Response
Thank you for your replies.
Dear Reviewer, thank you for accepting our answers.
The authors explained to me the difference of this manuscript with their previous work [34] (C11), but they didn't account for this difference and the novelty in the modified version of the manuscript.
The novelty of our work in comparison to the earlier work is as follows:
- In our earlier work, now cited as [41], we did not apply an important preprocessing step that allows automatic identification of the Region of Interest (RoI), i.e., in this work we applied the total variance map defined by equation (1) to this task
- In previous work, the proposed method was tested on only a few LHSV recordings, while the current work includes a more complete study conducted for 22 cases. We also show that the automatic segmentation method works for both normophonic and dysphonic voices.
- Finally, our results were positively validated by phoniatricians, which was not the case in the earlier study
We have added additional comments in the manuscript where we cite our previous work.
We also believe that the presented method, in which the image segmentation procedure is optimized using an acoustic signal, will be of interest to a wider community of Sensors readers.
C12: Thank you for providing these data.
Thank you
C25, C26: Thanks for providing these methodological details useful for the reproducibility.
Thank you
C28: Thank you for addressing this rather crucial point.
Thank you
Again, we are grateful for your extensive comments, which allowed us to significantly improve the manuscript.